# Application of BiOX Photocatalyst to Activate Peroxydisulfate Ion-Investigation of a Combined Process for the Removal of Organic Pollutants from Water

**Tünde Alapi** [1,*] **, Bence Veres** [1] **, Máté Náfrádi** [1] **, Luca Farkas** [1] **, Zsolt Pap** [2] **and Anett Covic** [1]

1    Department of Inorganic and Analytical Chemistry, University of Szeged, Dóm Square 7,
     H-6720 Szeged, Hungary
2    Applied & Environmental Chemistry Department, University of Szeged, Rerrich Béla Square 1,
     H-6720 Szeged, Hungary
*    Correspondence: alapi@chem.u-szeged.hu

**Abstract:** The persulfate-based advanced oxidation processes employing heterogeneous photocatalysts to generate sulfate radicals ($SO_4^{\bullet-}$) from peroxydisulfate ion (PDS, $S_2O_8^{2-}$) have been extensively investigated to remove organic pollutants. In this work, BiOX (X = Cl, Br, and I) photocatalysts were investigated to activate PDS and enhance the transformation rate of various organic substances under UV (398 nm) and Vis (400–700 nm) radiation. For BiOCl and BiOBr, in addition to excitability, the light-induced oxygen vacancies are decisive in the activity. Although without organic substances, the BiOI efficiency highly exceeds that of BiOBr and BiOCl for PDS activation (for BiOI, 15–20%, while for BiOBr and BiOCl, only 3–4% of the PDS transformed); each BiOX catalyst showed enhanced activity for 1,4-hydroquinone (HQ) transformation due to the semiquinone radical-initiated PDS activation. For sulfamethoxypyridazine (SMP), the transformation is driven by direct charge transfer, and the effect of PDS was less manifested. BiOI proved efficient for transforming various organic substances even under Vis radiation. The efficiency was enhanced by PDS addition (HQ is wholly transformed within 20 min, and SMP conversion increased from 40% to 90%) without damaging the catalyst; its activity did change over three consecutive cycles. Results related to the well-adsorbed trimethoprim (TRIM) and application of biologically treated domestic wastewater as a matrix highlighted the limiting factors of the method and visible light active photocatalyst, BiOI.

**Keywords:** benzoquinone; sulfonamides; sulfate radical ion; semiquinone; matrix effect; BiOX

## 1. Introduction

Nowadays, environmental pollution, including water pollution, has received particular attention, and treating water pollutants has become an urgent task. Some biologically active organic contaminants, such as endocrine-disrupting chemicals, pharmaceuticals, and personal care products, are non-biodegradable and cannot be removed entirely using conventional water treatment methods. Therefore, the effective elimination of these contaminants has become a leading research topic in environmental sciences. Moreover, the world has faced the pandemic caused by COVID-19, which resulted in the growing global use of antibiotics and their concentration in wastewater [1–3]. The concentration of antibiotics in influent wastewater is typically in the order of nanograms per liter and often reaches several milligrams per liter. During wastewater treatment, their concentration is usually reduced, although a significant part of them and some of their metabolites can often remain in the effluent and reach the surface waters. This further increases the probability of developing antibiotic-resistant bacterial strains, having unpredictable consequences [4].

Heterogeneous photocatalysis is one of the Advanced Oxidation processes (AOP), which is suitable for the economical removal of non-biodegradable water pollutants, especially by using photocatalysts that can be excited by visible light. In a photocatalyst excited,

the electron ($e_{cb}^-$) in the conduction band and hole ($h_{vb}^-$) in the valence band, created by photoinitiated charge separation, must avoid recombination and reach the surface to react with the organic compound or generate reactive species (radicals and radical ions) to degrade it. The widely studied first-generation photocatalysts, such as $TiO_2$ and ZnO, can be excited in the UV range, and the transformation of the organic substances is generally initiated by hydroxyl radical ($^\bullet$OH) based reactions [5]. In recent decades, the number of publications related to synthesizing and testing visible light-excitable photocatalysts with lower bandgap energy has increased significantly [6–8]. Another important goal is to achieve a longer lifetime of the photogenerated charges to enhance the oxidation and reduction processes realized through direct charge transfer on the surface and increase the efficiency of the transformation of organic and inorganic components in this way [5,9].

Bismuth-containing photocatalysts are intensively researched materials; most publications focus on bismuth oxohalides (BiOX, X = F, Cl, Br, I) due to their highly unique structure and beneficial optical and catalytic properties. Furthermore, their advantages are chemical stability, easy preparation, good adsorption properties, and superior photocatalytic activity [10–15]. BiOF and BiOCl are active in the UV range (band gap: ~3.6 eV for BiOF and ~3.2 eV for BiOCl), while BiOBr, and especially BiOI, are active in the visible range (band gap: ~2.6 eV for BiOBr and ~1.8–2.1 eV for BiOI) [14,16]. In addition to optical properties and widely different bandgap values, the adsorption capacity and surface properties of BiOX photocatalysts also differ. Photoinduced oxygen vacancies (OVs) are easily formed on their surface (especially for BiOBr and BiOCl), which can change their color, absorption, and adsorption properties. All of these can be beneficial in terms of photocatalytic activity [17,18]. The number and quality of reactive species formed on the surface of the excited catalyst are decisive for the efficiency and transformation of the organic target substances. The relative oxidation power of photogenerated $h_{VB}^+$ changes in the order of BiOCl > BiOBr > BiOI. The $^\bullet$OH forms with low efficiency in the case of BiOBr, while the $O_2^{\bullet-}$ formation is characteristic for each BiOX [19]. Several attempts have been made to enhance the efficiency of BiOX photocatalysts, including the preparation of composite catalysts, such as BiOCl/BiOBr [20], BiOCl/BiOI, [21–23], and $TiO_2$/BiOX [24], with improved stability and enhanced visible-light activity.

The efficiency of heterogeneous photocatalysis can be enhanced via the addition of substances that are effective electron scavengers and the source of highly reactive radicals simultaneously, such as ozone [25] and $H_2O_2$ [26]. In recent years, persulfate (PS)-based AOPs have been considered a promising method to eliminate organic pollutants in water due to the favorable properties of sulfate radical ion ($SO_4^{\bullet-}$), which is a strong nonselective oxidant with high redox potential (2.6–3.1 V), similar to the $^\bullet$OH (1.9–2.7 V) [27–32]. The source of $SO_4^{\bullet-}$ can be peroxomonosulfate ($HSO_5^-$, PMS) or peroxydisulfate ion ($S_2O_8^{2-}$, PDS); their activation can occur via energy or electron transfer. In the case of the combination PS process with heterogeneous photocatalysis, the excited photocatalysts act as electron donors, and PS can capture photogenerated $e_{cb}^-$ to generate $SO_4^{\bullet-}$. In addition to $SO_4^{\bullet-}$, other reactive species, such as singlet oxygen ($^1O_2$), also form and are responsible for the enhanced transformation and mineralization rate of target substances. Simultaneously, the photocatalytic process can be promoted due to the improved separation of charge carriers.

Recently intensive studies were conducted to investigate the combination of heterogeneous photocatalysis with the PS process [33–37], using PDS [38,39] and PMS [40–42] to enhance the efficiency. Some publications have reported the efficient application of BiOBr [43] or BiOI composite to activate PDS [44–46]. However, to our knowledge, the direct comparison of pure BiOI, BiOBr, and BiOCl for activation of PDS under UV and visible light has not yet been published. In this work, we compare the efficiency of the BiOX photocatalysts for the activation of PDS and the conversion of different organic compounds (1,4-benzoquinone (BQ), 1,4-hydroquinone (HQ), sulfamethoxypyridazine (SMP), and trimethoprim (TRIM)). The reusability of the BiOX catalyst showing the best activity and the investigation of its use in biologically treated municipal wastewater as a matrix was also studied.

## 2. Results and Discussion

### 2.1. Characterization of the BiOX Photocatalyst

The synthesis method optimized and published by Bárdos et al. [15] was used to prepare the BiOX photocatalysts, having spherical hierarchical microcrystals composed of individual nanoplates. The synthesized catalysts were characterized using powder X-ray diffractometry (XRD). The specific surface area (using measurements based on N$_2$ adsorption), the reflectance, and bandgap values (using DRS measurements) were determined. The results (Figure 1 and Table 1) provided data consistent with the data in the literature [13,47]. According to the bandgap value, BiOBr can be excited with the photons emitted by UV-LED, while BiOI can be excited by both LEDs. However, a significant part of the photons ($\lambda > 512$ nm) emitted by the Vis-LED is unsuitable for creating the photogenerated e$_{cb}^-$–h$_{vb}^+$ pair. The energy of 398 nm photons is inadequate for the excitation of BiOCl, having a 3.26 eV bandgap value.

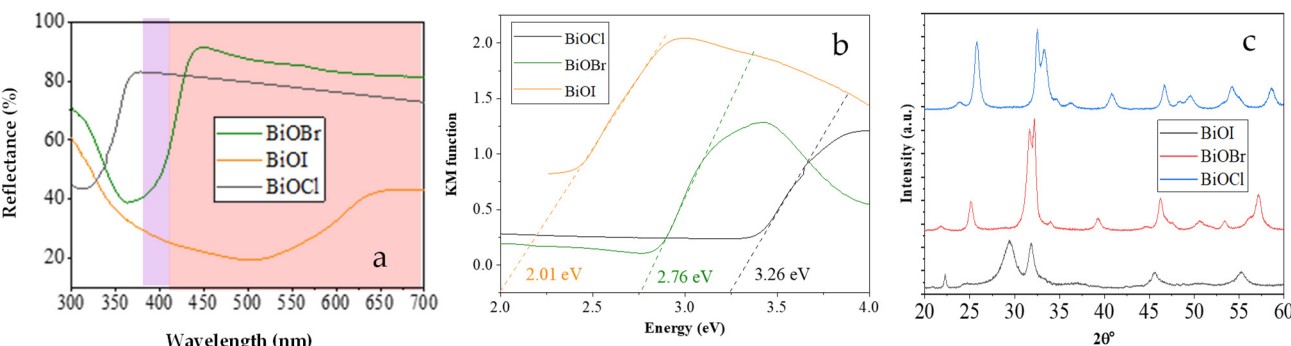

**Figure 1.** UV–Vis DRS spectra (**a**) (shadow indicates the emission range of LED (violet: UV-LED, red: Vis-LED)), Kubelka Munk Curves (**b**), and XRD patterns (**c**) of BiOCl, BiOBr, and BiOI.

**Table 1.** The specific surface areas and bandgap values of the photocatalysts.

|  | Bandgap | Surface Area | Primary Particle Size |
|---|---|---|---|
| BiOCl | 3.26 eV (363 nm) | 16 m$^2$ g$^{-1}$ | 31 nm |
| BiOBr | 2.76 eV (435 nm) | 18 m$^2$ g$^{-1}$ | 39 nm |
| BiOI | 2.01 eV (512 nm) | 45 m$^2$ g$^{-1}$ | 21 nm |

### 2.2. Transformation of the Target Substances Using BiOX Photocatalysts

The efficiency of the photocatalytic elimination of organic substances depends on the quality and quantity of the reactive species formed on the surface of the excited photocatalyst and the reactivity of the target substances towards them. The formation of the non-selective $^\bullet$OH is unlikely in the case of the irradiated BiOX [13,19,48]; the transformation of organic substances occurs mainly via reactions with other, more selective species, such as singlet oxygen ($^1$O$_2$) or O$_2^{\bullet-}$/HO$_2^\bullet$ [19]. Another option for the transformation of organic substances is the direct charge transfer [13,48]. In this work, four organic substances were chosen to determine and compare the activity of BiOX photocatalyst and investigate the relative contribution of the various processes to the transformation of organic substances. The target compounds were carefully chosen to characterize the activity of catalysts from different points of view.

#### 2.2.1. Transformation of BQ

Fónagy et al. [49] demonstrated that the transformation of BQ in O$_2$-free suspension occurs via direct charge transfer; the reaction with e$_{CB}^-$ results in the formation of HQ [50]:

$$BQ + 2\,e^- \rightarrow HQ \quad k = 3.9 \times 10^{10}\ M^{-1}\ s^{-1} \tag{1}$$

consequently, the transformation rate of BQ and formation rate of HQ is primarily determined by the density of the photogenerated $e_{cb}{}^-$ on the surface, which depends on the absorption property, excitability of the catalyst, and the recombination rate of photogenerated charges at a given photon flux and wavelength.

Figure 2 shows the BQ transformation in $O_2$-free suspensions of BiOX photocatalysts using UV LED (398 nm) and Vis LED (400–700 nm). BQ is also efficiently transformed by direct photolysis, even under Vis radiation. Although the transformation of BQ by both direct charge transfer and photolysis results in HQ, there is a characteristic difference between the two processes regarding HQ conversion. In the case of direct charge transfer, BQ completely transforms to HQ [49], while in the case of direct photolysis besides HQ, other aromatic and ring-opening products form [51].

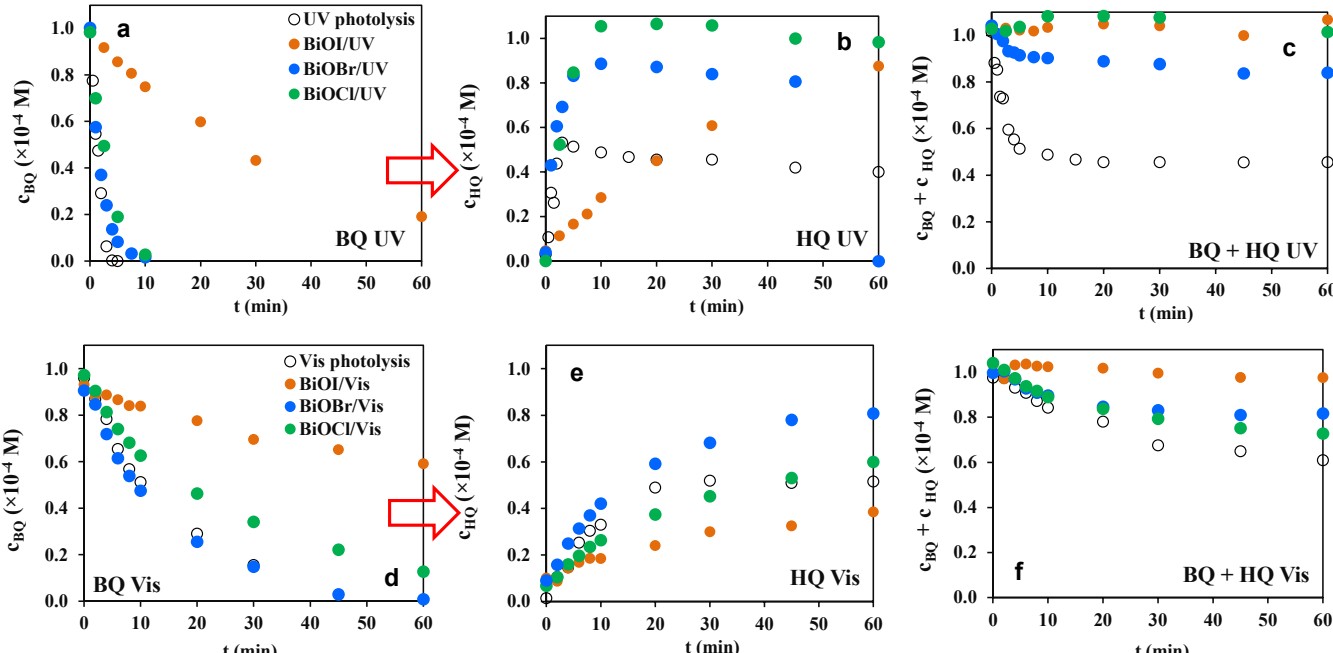

**Figure 2.** Transformation of BQ (**a**,**d**) and formation of HQ (**b**,**e**) and the sum of the BQ and HQ concentration (**c**,**f**) in $O_2$-free BiOX suspensions, under UV (**a**–**c**) and Vis (**d**–**f**) radiation (**a**,**d**).

Based on the bandgap energy (Table 1), BiOI and BiOBr can be excited with 398 nm photons, but not BiOCl, while only BiOI can be excited using a Vis LED. Nevertheless, in the case of UV LED, the transformation of BQ occurred at a rate similar to that of photolysis in the case of BiOCl and BiOBr, and transformation was observed even under visible light radiation. Both catalysts were blackened under irradiation; this change was reversible, and both photocatalysts changed their color back in the dark. A similar observation has been published and interpreted with photoinitiated oxygen/halogen vacancies, which may result in better absorption properties, band gap changes, and enhanced photocatalytic activity [18]. The oxygen vacancy-mediated mechanism of $CO_2$ photoreduction was suggested for BiOX [17], especially for BiOBr. In the case of BiOCl and BiOBr under UV light, the conversion of HQ from BQ was significantly higher (85 and 100%) than in the case of photolysis (less than 50%), which suggests that the heterogeneous photocatalytic transformation contributes to the BQ transformation (Figure 2). However, the direct photolysis of BQ must be addressed, especially for BiOCl and BiOBr. Within the emission range of the Vis LED, BiOCl, and BiOBr show high reflectance, and HQ conversion is closer to that measured during photolysis, indicating that it contributes significantly to the conversion.

The BiOI catalyst should be discussed separately. The BiOI can be excited by both LEDs and, opposite to the BiOCl and BiOI, efficiently absorb the emitted photons even in the emission range of Vis LED (Figure 1 and Table 1). The color change of BiOI was not observed under irradiation. The transformation of BQ in BiOI suspension was much

slower than in solution via photolysis, and BQ completely transformed into HQ (Figure 2). Presumably, in the case of BiOI suspension, the photons are absorbed by the catalyst and the dominant way of BQ transformation is the direct charge transfer, beside that the role of photolysis is negligible.

### 2.2.2. Transformation of HQ

The HQ transformation was investigated in aerated suspensions (Figure 3). The concentration of the HQ does not change in irradiated and aerated solutions or $O_2$-free suspensions and is poorly (<2%) adsorbed on the surface of photocatalysts. The transformation requires the simultaneous presence of BiOX photocatalyst, irradiation, and dissolved $O_2$ [50] and occurs due to the reaction with $O_2^{\bullet-}$ [52] resulting in BQ. In this case, the HQ transformation rate directly refers to the $O_2^{\bullet-}$ formation rate [49].

$$O_2 + e^- \rightarrow O_2^{\bullet-} \qquad\qquad k = 2.3 \times 10^{10}\ \mathrm{M^{-1}\ s^{-1}} \tag{2}$$

$$HQ + O_2^{\bullet-} \rightarrow HBQ^{\bullet} + HO_2^- \tag{3}$$

$$HBQ^{\bullet} \rightarrow BQ^{\bullet-} + H^+ \tag{4}$$

$$BQ^{\bullet-} + O_2 \rightarrow BQ + O_2^{\bullet-} \tag{5}$$

$$BQ + O_2^{\bullet-} \rightarrow BQ^{\bullet-} + O_2 \qquad k = 9.8 \times 10^8\ \mathrm{M^{-1}\ s^{-1}} \tag{6}$$

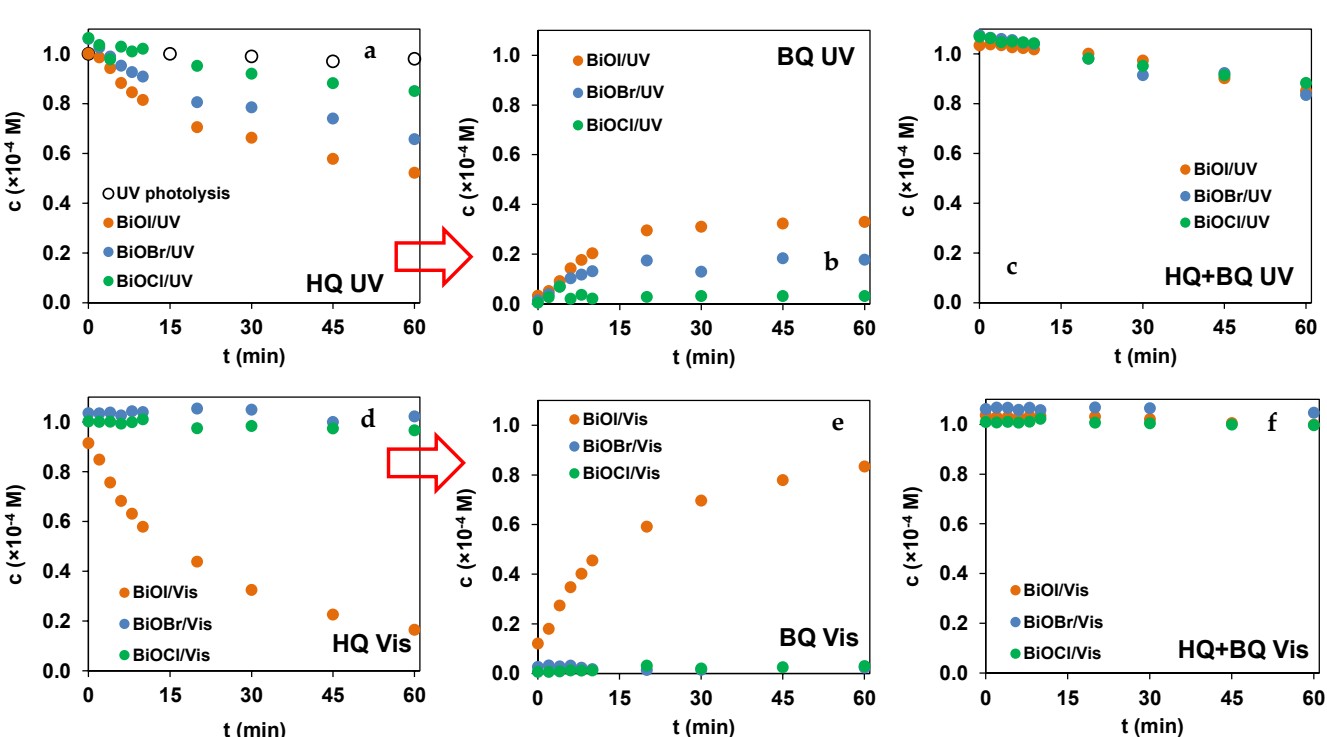

**Figure 3.** Transformation of HQ (**a**,**d**) and formation of BQ (**b**,**e**) and the sum of the HQ and BQ concentration (**c**,**f**) in aerated BiOX suspensions under UV (**a**–**c**) and Vis (**d**–**f**) radiation.

The formation rate of $O_2^{\bullet-}$ determined by Lv et al. [19] under simulated sunlight and changed in the order BiOCl > BiOBr $\cong$ BiOI, while the relative oxidation power of $h_{vb}^+$ was in the order of BiOCl > BiOBr > BiOI. However, in our case, the transformation rate of HQ changed in the order BiOI > BiOBr > BiOCl, reflecting the optical properties and excitability of the catalysts (Figures 1 and 3, Table 1). Under Vis radiation, the activity of BiOCl and BiOBr was negligible, while more than 80% of HQ was converted to BQ within 60 min in the BiOI suspension. The results are consistent with the bandgap values, based on which BiOCl and BiOBr cannot be excited by visible light, or only to a small extent, while BiOI

can be well excited even by Vis light. The blackening of BiOCl and BiOBr was observed in suspensions containing $O_2$ and was less intense under Vis radiation than under UV. The sum of the HQ and BQ concentration decreased slowly in UV, while did not change in Vis irradiated suspensions (Figure 3). The slight decrease in the sum of HQ and BQ concentration suggests that further BQ transformation occurs, resulting in the ring-opening process. For BiOI, the application of Vis LED proved to be more efficient than UV LED at the same electric power (4.6 W), even though the UV photon flux is twice the photon flux in the visible range.

### 2.2.3. Transformation of SMP and TRIM

SMP and TRIM, as widely used, hardly biodegradable antibiotics, often detected in surface water, were used as environmentally relevant target substances. A significant difference between the substrates is that SMP is poorly adsorbed on BiOX photocatalysts, while the adsorption of TRIM is substantial, particularly in the case of BiOI.

The SMP transformation using BiOI photocatalysts was investigated by Náfrádi et al. [48], and the direct charge transfer was reported as the primary transformation way. The indirect photodegradation [53,54] and reaction with $e_{cb}^-$ cannot be excluded partly because of the reactivity of pyridazine moiety towards $e_{aq}^-$. However, in the case of $TiO_2$ excited with 398 nm light, direct energy transfer [55] was observed, and a reaction with $^1O_2$ [56] was also assumed. Similar to HQ, direct photolysis of SMP is negligible in $O_2$-free irradiated suspension. The transformation surprisingly occurred at similar rates for all three catalysts using UV LED and was much faster than HQ transformation under the same experimental conditions (Figure 4). The observed activity cannot be interpreted solely based on excitability, especially in the case of BiOCl [57]. Most probably, the photoinitiated change of the catalysts properties, the formed oxygen vacancies, and the surficial $Bi^{5+}$ and $Bi^0$ play a significant role in the efficiency and highly favorable to the redox capacity for the photocatalysts and transformation of SMP by direct charge transfer. The synergism between $Bi^0$ and $Bi^{5+}$ was demonstrated and was proved to be the reason for the enhanced degradation of organic substances [52–55]. The decomposition was a hole-driven process; moreover, $O_2^{\bullet-}$ has an additive role. However, under visible light illumination, only the BiOI catalyst showed activity; the transformation efficiency was half that observed under UV radiation (Figure 4), similar to the BQ transformation also driven by direct charge transfer (Figure 2).

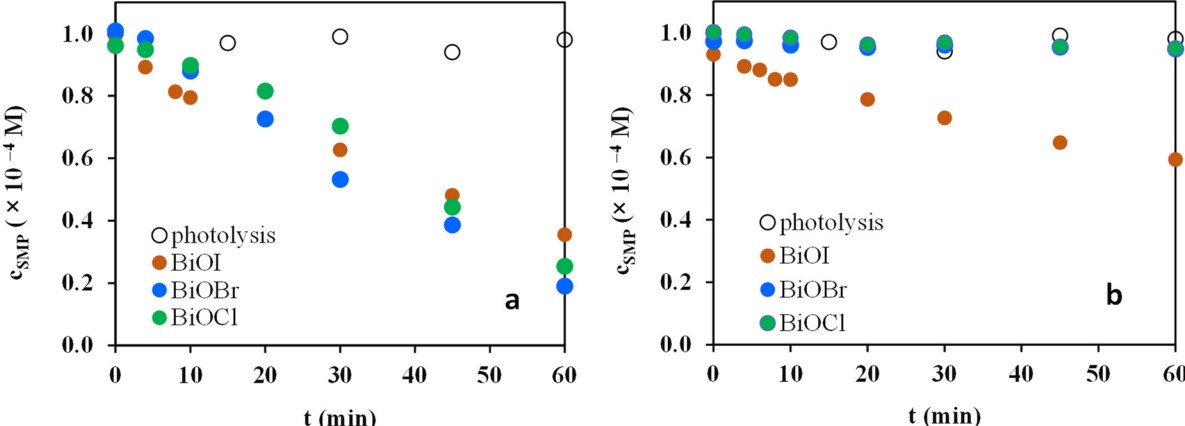

**Figure 4.** Transformation of SMP in aerated BiOX suspensions under UV (**a**) and Vis (**b**) irradiation.

Opposite the HQ and SMP, the relative adsorbed amount of TRIM was 39% for BiOI. Lower values, 6% for BiOBr, and less than 2% for BiOCl were determined. To distinguish the amount of adsorbed and transformed TRIM; NaF solution was added to the samples for TRIM desorption before analysis [48]. Opposite to the good adsorption properties, no more than 6% of TRIM transformed in BiOX-containing suspension, even in the case of

BiOI (Figure S1). The reason could be the low reactivity of TRIM towards less reactive ROS, such as $O_2^{\bullet-}$ and $^1O_2$. The transformation by charge transfer is most probably ruled out in this case. This result clearly shows the limitation of the applicability of BiOX photocatalysts because of the lack of the effective formation of highly oxidizing non-selective species, such as $^\bullet OH$.

### 2.3. Effect of PDS

The effects of PDS on the BiOX Structure and Properties, and the Transformation of PDS without Organic Substances

The effect of $1.0 \times 10^{-3}$ M PDS on the photocatalyst stability was investigated under UV and Vis radiation. XRD and DRS results proved that there is no significant change in their structure, optical properties, and excitability of the photocatalysts (DRS spectra and band gap values have not changed) (Figure 5).

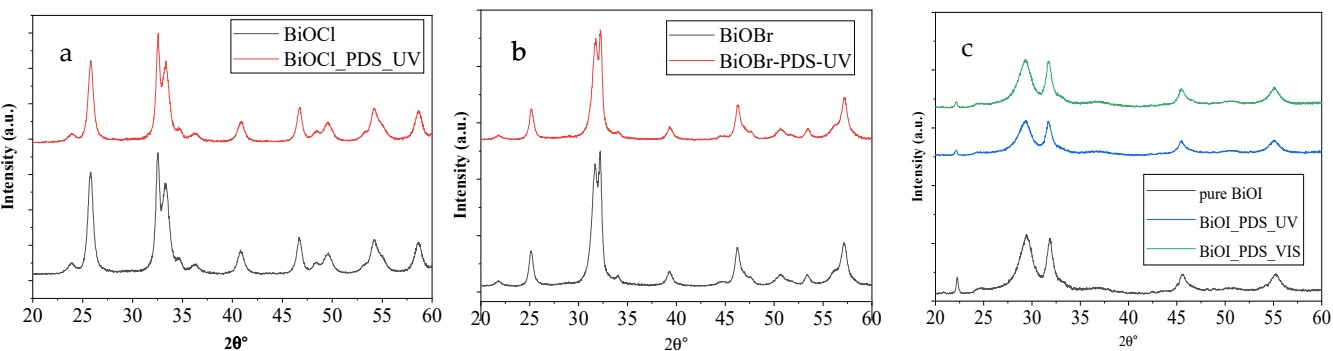

**Figure 5.** The Effect of PDS on the XRD diffractogram of BiOX ((**a**): BiOCl; (**b**): BiOBr; (**c**): BiOI) photocatalysts under UV and Vis radiation in aerated suspensions.

In an aerated BiOI suspension containing no organic matter, the transformation of $1.0 \times 10^{-3}$ M PDS results in the formation of $3.0 \times 10^{-4}$ M and $3.7 \times 10^{-4}$ M $SO_4^{2-}$ under UV and Vis radiation, respectively, i.e., 15–20% of PDS was transformed in 60 min. However, in the case of BiOBr and BiOCl, only 3–4% PDS was converted even under UV radiation (Figure 6). That means BiOI is much more efficient for PDS reduction than BiOBr or BiOCl.

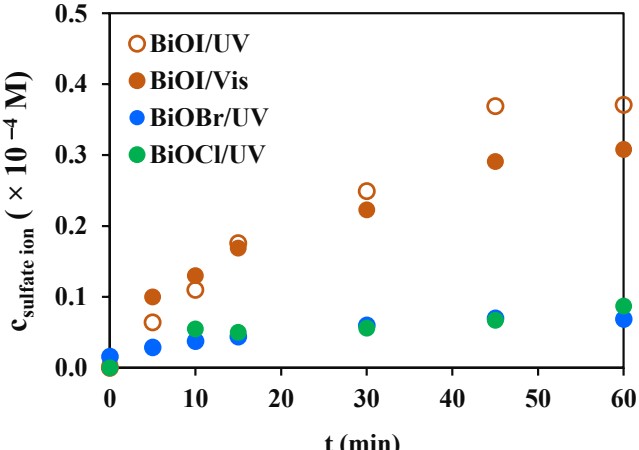

**Figure 6.** The concentration of sulfate ion results in PDS ($1.0 \times 10^{-3}$ M) transformation in UV and Vis irradiated, aerated BiOX suspensions without organic substances.

The ineffectiveness of BiOCl is not surprising given its excitation and absorption properties (Table 1); however, better efficiency of UV-irradiated BiOBr was expected, as publications reported for activation of PDS [43,58] or PMS [59], even under visible light



radiation. In the case of PDS, the enhanced Bisphenol-A transformation was interpreted by oxygen vacancies mediated PDS activation process resulting in $^1O_2$ as the main reactive species under alkaline suspension of BiOBr [43]. A likely explanation of our results could be the competition between $O_2$ and PDS for $e_{cb}^-$. The $e_{cb}^-$ capture by $O_2$ can be more favorable for BiOBr than BiOI due to the different conduction band potentials [19]; thus, $O_2$ can effectively compete with PDS for $e_{cb}^-$. Another important factor could be the surface charge, which is negative for all BiOX to varying degrees depending on the halogen atom [22,60] and probably less favorable for the surface reactions of negatively charged ions, such as PDS for BiOCl and BiOBr than BiOI. The different surface properties of BiOX were also well reflected in the amount of adsorbed SMP.

### 2.4. Effect of PDS on the Transformation of the Target Substances

Without irradiation, the organic substances do not transform; the direct oxidation with PDS is negligible in each case. The positive effect of PDS was pronounced in the case of HQ not only for BiOI but also for BiOCl and BiOBr photocatalysts (Figure 7); however, the PDS transformation was efficient without an organic substrate only for BiOI (Figure 6). In the case of BiOI, the enhanced transformation is most likely caused by the formed $SO_4^{•-}$. However, this interpretation needs to be revised for BiOBr and BiOCl.

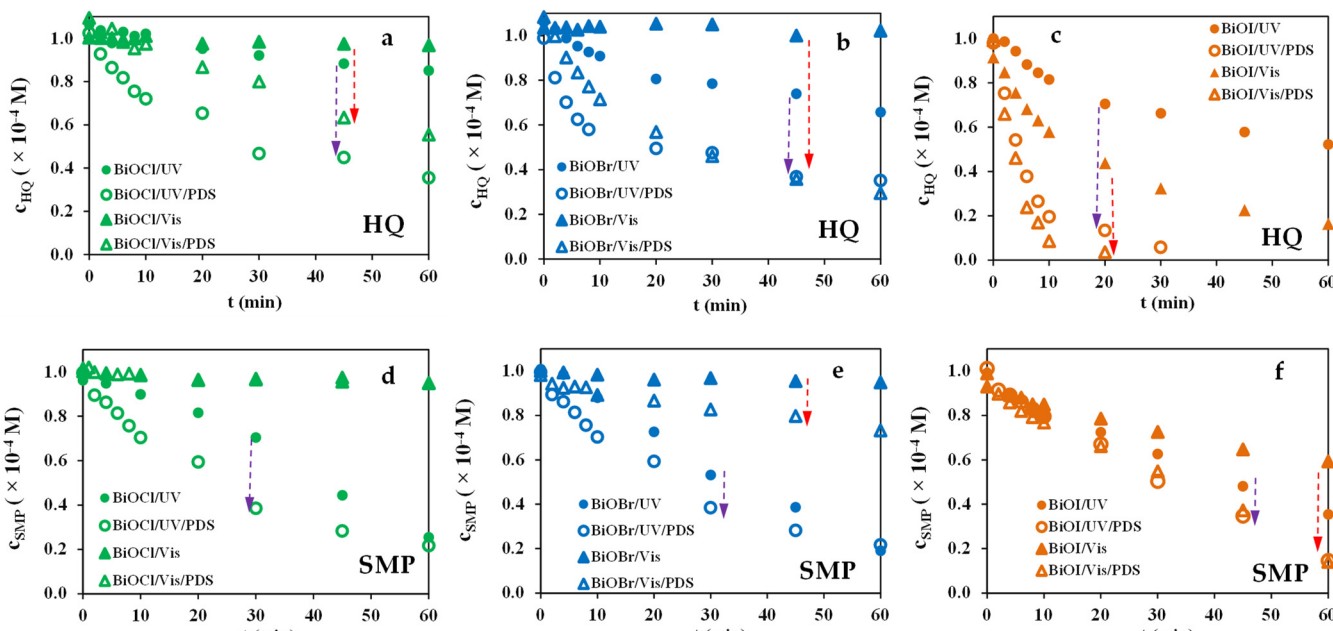

**Figure 7.** The effect of $1.0 \times 10^{-3}$ M PDS on the transformation of HQ (**a–c**) and SMP (**d–f**) in UV and Vis irradiated, aerated BiOCl (**a,d**), BiOBr (**b,e**), and BiOI (**c,f**) suspensions. (The narrows show the effect of PDS (purple: UV; red: Vis).

The transformation of HQ results in BQ via semiquinone radical (HBQ$^•$), which also can form via the comproportionation between BQ and HQ. The PDS can undergo reductive transformation into $SO_4^{•-}$ by HBQ$^•$ and enhance the transformation rate in this way [61–63]. However, the authors interpret the transformation of various organic substances in the presence of PDS or PMS differently. Fang et al. [62] reported that the $SO_4^{•-}$ is responsible for the enhanced 2,4,4′-trichlorobiphenyl transformation in the case of HBQ$^•$ activated PDS. Similarly, Zhou et al. [61] activated PMS by HBQ$^•$ for the degradation of sulfamethoxazole. The $^•OH$ or $SO_4^{•-}$ were not detected, and $^1O_2$ was found to be responsible for sulfamethoxazole transformation. Similarly, Bu et al. [43] reported that the photon-initiated oxygen vacancies, $Bi^{5+}$ and $Bi^0$, are responsible for the PDS transformation and cause enhanced activity of BiOBr in the presence of PDS. The Bisphenol-A

transformation was justified by the formation of $^1O_2$, but the contribution of $SO_4^{\bullet-}$ was not confirmed.

Comparing the effect of PDS on HQ degradation for BiOX catalysts, BiOI is the most advantageous; the HQ is wholly transformed within 20 min. For BiOBr and BiOCl, 75% conversion was reached during 60 min even under UV radiation. Their efficiency was far exceeded by BiOI, even under Vis radiation. Most likely, in the case of BiOCl and BiOBr, mainly $HBQ^{\bullet}$ initiated the transformation of PDS is responsible for the enhanced efficiency. In addition, the photon-initiated oxygen vacancies, the surficial $Bi^{5+}$ and $Bi^0$ may have a role. However, estimating the contribution of individual effects is beyond the scope of our work and requires several additional measurements.

The effect of PDS on SMP conversion is moderated compared to that observed for HQ (Figure 7), where the $HBQ^{\bullet}$-induced transformation of PDS also contributes. For BiOCl and BiOBr catalysts, the increased efficiency was observed only under UV radiation, confirming the crucial role of oxygen vacancies in activating PDS. The SMP transformation was enhanced at the beginning of the treatment; after that, the transformation slowed down. For BiOI, the positive effect of PDS was pronounced under UV and Vis radiation. After 60 min of Vis irradiation, the 40% conversion increased to more than 90% (Figure 7), while the SMP conversion increased by 10% under UV radiation. However, the conversion efficiency of SMP was still much lower than that of HQ when the $HBQ^{\bullet}$-initiated process also contributed to the enhanced efficiency.

The effect of PDS on TRIM transformation was investigated using BiOI under Vis radiation, and a slight increase in efficiency was observed; nearly 24% of TRIM was transformed (Figure S1). However, this measurement also confirms that the activity of BiOX catalysts is highly dependent on the compound to be converted, even in the presence of PDS. Moreover, the well-adsorbed TRIM can partially inhibit PDS activation on the surface.

Although $SO_4^{\bullet-}$ is effective for mineralization, significant TOC changes were not observed during the treatments. TOC did not decrease during the 60-min treatment without PDS, while a maximum decrease of 10% was measured in the presence of PDS for HQ and SMP. The pH of BiOX suspension was about 6.0–6.5, the addition of target substances did not significantly change that. During treatment, the pH decreased slightly (to 5.0–4.0 without PDS, and to 4.5–3.7 in the presence of PDS), likely because of the formed organic acids. Probably, pH change has no significant effect on PDS transformation and $SO_4^{\bullet-}$ initiated reactions [64]; however, it can affect the surface properties of the photocatalysts and the $HO_2/O_2^{\bullet-}$ ratio during treatment.

### 2.5. Effect of Dissolved Oxygen

PDS can affect the conversion rate of organic compounds in heterogeneous photocatalysis in many ways, such as forming new reactive species ($SO_4^{\bullet-}$, $^1O_2$) and enhancing the charge separation due to the inhibition of the recombination of photogenerated charges [34,36,65]. To investigate this phenomenon, measurements were performed in $O_2$-free suspensions (Figure 8).

While the conversion rates of BQ and SMP in $O_2$-free suspensions without PDS were negligible, the degradation proceeded efficiently in the presence of PDS (Figure 8). A significant difference is that in the case of HQ, the positive effect of dissolved $O_2$ is well manifested even in the presence of PDS. In contrast, for SMP, the transformation rate is not affected by $O_2$. In the case of HQ, $O_2$ plays an important role, as $O_2^{\bullet-}$ is highly efficient in its conversion, while for SMP, the direct charge transfer was assumed as the primary process. The PDS probably partially takes over the electron capture role of $O_2$ in the case of BiOI. Still, the positive effect of the formed $SO_4^{\bullet-}$ overcompensates this adverse effect, and both reactive species can contribute significantly to the HQ transformation in aerated suspension. In the case of SMP, the lack of an $O_2$ effect suggests that there is no contribution of $^1O_2$ or $O_2^{\bullet-}$ to the transformation in the presence of PDS and confirms the additive role of the highly reactive $SO_4^{\bullet-}$, the formation of which is not related to the dissolved $O_2$.

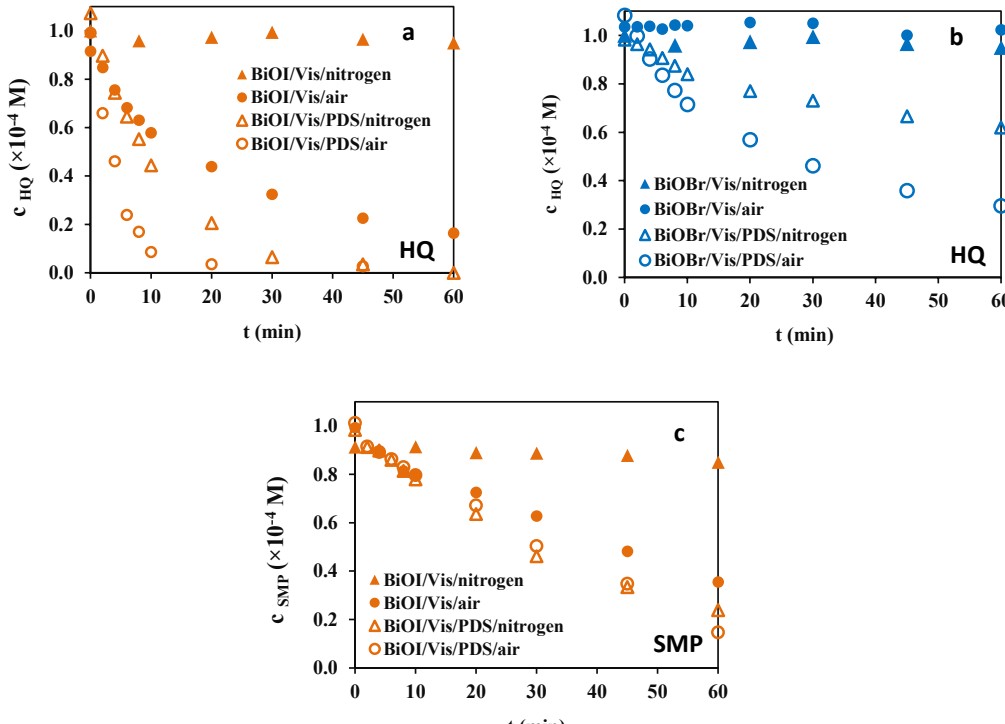

**Figure 8.** The Effect of dissolved $O_2$ on the transformation of HQ (**a**,**b**) and SMP (**c**).

### 2.6. Reusability of the BiOI Photocatalyst

The study of the stability and reusability of the catalyst has a crucial role in practical application. BiOI, the most efficient photocatalyst under Vis radiation, was chosen to investigate reusability. Another reason was that the efficient transformation of PDS without organic substances was proved only in the case of BiOI.

Degradation of $1.0 \times 10{-}4$ M HQ and $5.0 \times 10{-}5$ M SMP was monitored for three consecutive cycles (Figure 9), under Vis radiation, using $5.0 \times 10{-}4$ M PDS concentration. After the end of a cycle, the concentration was adjusted to $1.0 \times 10{-}4$ M for HQ and $5.0 \times 10{-}5$ M for SMP by adding a small volume of concentrated solution. $5.0 \times 10{-}4$ M PDS was added to the suspension at the beginning of each run.

The transformation rate of both target substances did not change during the three consecutive cycles (Figure 9). Slightly decreased activity may be due to products' accumulation and competition with HQ and SMP for reactive species. There is no significant change in the XRD patterns and DRS spectra after the third cycle (Figures S2 and S3); thus, there is probably no change in the photocatalyst structure. Based on the ion chromatography measurements, the $SO_4^{2-}$ concentration at the end of the cycle was $2.77 \times 10^{-4}$ M, $5.23 \times 10^{-4}$ M, and $7.80 \times 10^{-4}$ M, so the extent of PDS decomposition was almost constant ($1.4 \times 10^{-4}$ M, $1.2 \times 10^{-4}$ M, and $1.3 \times 10^{-4}$ M PDS decomposed during the first, second and third cycle), confirming the unchanged activity of the BiOI photocatalyst.

### 2.7. Effect of Inorganic Ions and Biologically Treated Domestic Wastewater as Matrix

The effect of matrix and matrix components is important for practical applicability; thus, pH, $Cl^-$ ($3.4 \times 10^{-3}$ M), $HCO_3^-$ ($8.6 \times 10^{-3}$ M), and biologically treated domestic wastewater on the efficiency was investigated (Figure 10). The concentration of $Cl^-$ and $HCO_3^-$ was adjusted to their concentration in the biologically treated wastewater. Table S1 shows the parameters of the matrix.

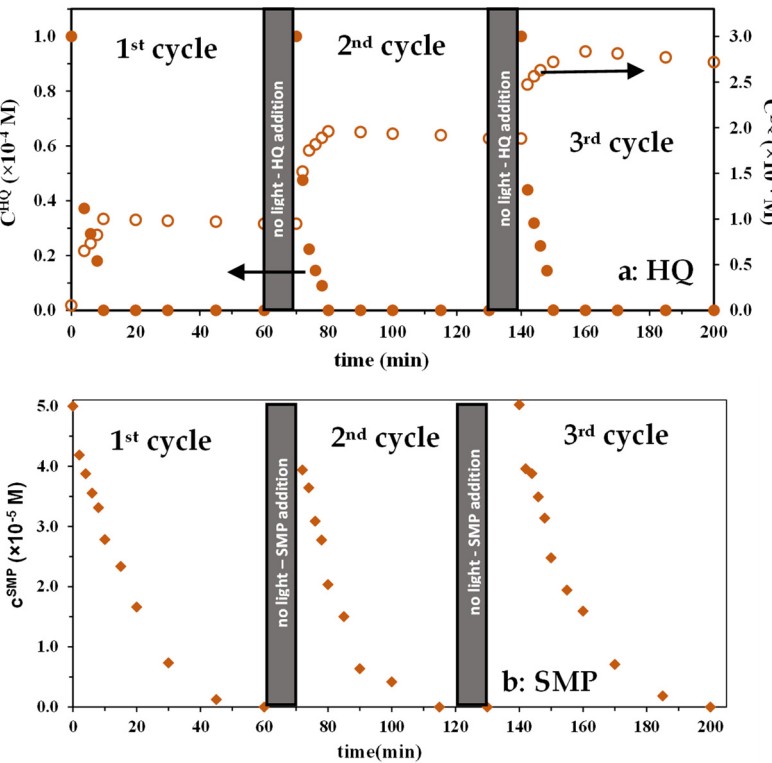

**Figure 9.** The transformation of HQ and formation of BQ (**a**) and the transformation of SMP (**b**) during three consecutive cycles in aerated BiOI suspensions, irradiated with Vis LED ($c_0$(HQ) = 1.0 × 10$^{-4}$ M; $c_0$(SMP) = 5.0 × 10$^{-5}$ M; $c_0$(PDS) = 5.0 × 10$^{-4}$ M).

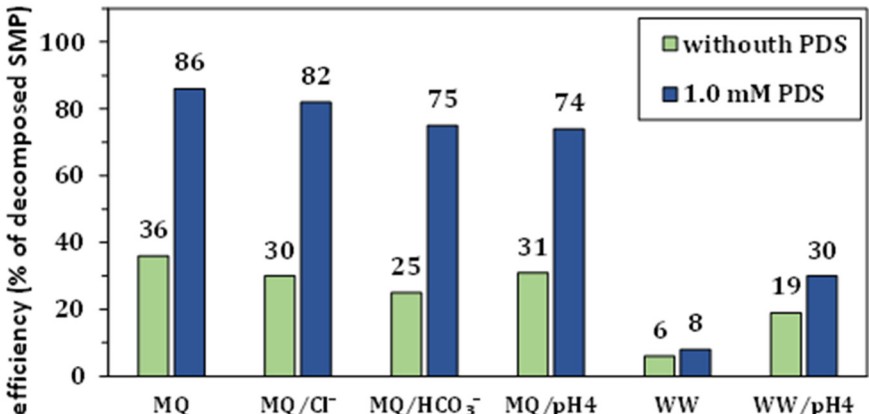

**Figure 10.** The effect of biologically treated domestic wastewater (ww), pH, Cl$^-$ (3.4 × 10$^{-3}$ M), and HCO$_3^-$ (8.6 × 10$^{-3}$ M) on the transformation of SMP ($c_0$(SMP) = 1.0 × 10$^{-4}$ M; $c_0$(PDS) = 1.0 × 10$^{-3}$ M) in BiOI suspension irradiated with Vis LED.

Biologically treated domestic wastewater almost wholly inhibited the transformation of both HQ and SMP, even in the presence of PDS. The matrix components may affect the surface properties of the photocatalyst, such as the surface charge and adsorption properties, or act as radical scavengers [66]. The effect of Cl$^-$ (3.4 × 10$^{-3}$ M) was negligible (Figure 10), even in PDS containing suspension; opposite that, Cl$^-$ can scavenge SO$_4$$^{\bullet-}$ (7) [67]. Nevertheless, the formed Cl-containing reactive species (8–10) [68] can contribute to the transformation of organic substances, and their further reactions can even result in $^\bullet$OH [69]. Consequently, the radical scavenging effect is not manifested in the decrease in the transformation rate [27].

$$Cl^- + SO_4^{\bullet-} \rightarrow Cl^\bullet + SO_4^{2-} \qquad\qquad k = 3.6 \times 10^8 \ M^{-1} \ s^{-1} \qquad\qquad (7)$$

$$Cl^{\bullet} + Cl^{-} \rightarrow Cl_2^{\bullet-} \qquad\qquad k = 7.8 \times 10^9 \ M^{-1} \ s^{-1} \qquad (8)$$

$$Cl^{\bullet}/Cl_2^{\bullet-} + H_2O \rightarrow ClOH^{\bullet-} + H^+/+ Cl^- \qquad (9)$$

$$ClOH^{\bullet-} \rightarrow {}^{\bullet}OH + Cl^- \qquad (10)$$

The $HCO_3^-$ ($8.6 \times 10^{-3}$ M) can also react with $SO_4^{\bullet-}$ [70] and results in $CO_3^{\bullet-}$ (11), a less reactive and selective species [71]:

$$HCO_3^- + SO_4^{\bullet-} \rightarrow CO_3^{\bullet-} + SO_4^{2-} + H^+ \qquad k= 2.8 \times 10^6 \ M^{-1} \ s^{-1} \qquad (11)$$

$HCO_3^-$ only slightly reduced the transformation rate (Figure 10), but the intensive orange color of the catalysts became white. XRD and DRS results proved that BiOI transforms into bismuth subcarbonate (($BiO)_2CO_3$) (Figures S2–S4). The band gap of the formed, white material was 3.12 eV, close to the value reported for $(BiO)_2CO_3$ [72]. Surprisingly, however, this was not reflected in the cessation of activity since $(BiO)_2CO_3$ is also photoactive [73,74]. The $Bi^{3+}$ entering the solution during the recrystallization could also activate the transformation of PDS. The harmful effect of $HCO_3^-$ can be eliminated by changing the pH to 4. The change of the pH to 4 with sulfuric acid caused a slight decrease in the transformation rate compared to that determined in Milli-Q water. In biologically treated domestic wastewater, the transformation rate increased due to the elimination of $HCO_3^-$ at acidic pH, but it was still significantly below the value measured in Milli-Q water (Figure 10). Probably the reason for this is not the effect of inorganic ions but organic compounds that are well adsorbed on the surface of BiOI. The TOC content of the wastewater decreased by 35% due to the addition of BiOI in the dark because of the adsorption of organic components. If they cannot transform, they block the transformation of the poorly adsorbed HQ, SMP, and PDS, and even the excitation of the BiOI. This assumption was confirmed by the observation; that the bleaching of BiOI in the matrix takes place much more slowly (Figure S4). This can also be interpreted by the adsorption of the organic components of the matrix on the surface, as they exert a protective effect against inorganic ions, i.e., the $HCO_3^-$.

## 3. Materials and Methods

### 3.1. Materials

Bismuth nitrate pentahydrate ($Bi(NO_3)_3 \times 5\ H_2O$), NaI, NaCl and NaBr, ethylene glycol, and ethanol were used for synthesizing photocatalysts. For photocatalytic test reactions, 1,4-hydroquinone (HQ), sulfamethoxypyridazine (SMP), and trimethoprim (TRIM) were used as target substances. Air or $N_2$ gas was applied to control the dissolved $O_2$ concentration of the treated solution or suspension. For investigation of the effect of peroxydisulfate ion, $Na_2S_2O_8$ (PDS) solution was added to the suspension. The pH was adjusted with $H_2SO_4$ or NaOH solutions. $Fe_2(SO_4)_3$, potassium oxalate, ammonium-reineckate salt, and 1,10-phenanthroline were used for the actinometric measurements. Table S2 shows the data of the materials used in this work. Each material was used without further purification. Table S1 shows the date of the biologically treated domestic wastewater (from the water treatment plant, Szeged, Hungary) used as a matrix.

### 3.2. Preparation of BiOX Photocatalysts

The BiOX photocatalysts were prepared via a solvothermal method, as described in the literature [13]. The $Bi(NO_3)_3 \times 5\ H_2O$ (2.07 g) and KI (0.7067 g), KBr (0.5066 g), or KCl (0.3174 g) were dissolved in 50 mL ethylene glycol with continuous stirring and heating (up to 45 °C). The suspension was heat-treated for 3 h at 120 °C in a PTFE-coated steel autoclave. The solid material was washed with distilled water and ethanol, then vacuum-filtered with a 0.1 μm pore size filter (Durapore®, hydrophilic PVDF) and dried for 24 h at 40 °C.

### 3.3. Photocatalytic Test Reactions

Two commercial LED tapes were used as light sources: the UV-LED emitting at 398 nm (LEDmaster, $\lambda_{emission}$= 398 ± 10 nm, 288 lumens, 4.6 W) and Vis-LED emitting

warm white light (LEDmaster, $\lambda_{\text{emission}}$= 400–700 nm, 600 lumens, 4.6 W). 1.0 m of LED tape (60 LED/meter) was fixed on the inside of the aluminum, double-walled reactor having a 66 mm inner diameter (Figure 11a,b). The reactors were equipped with a water-cooling system to ensure the LEDs' constant light output. The electrical power required to operate the LEDs was the same (4.6 W) in all cases; thus, the efficiency of the photocatalysts was determined at the same electrical energy input.

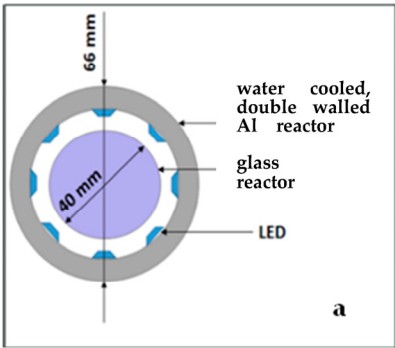
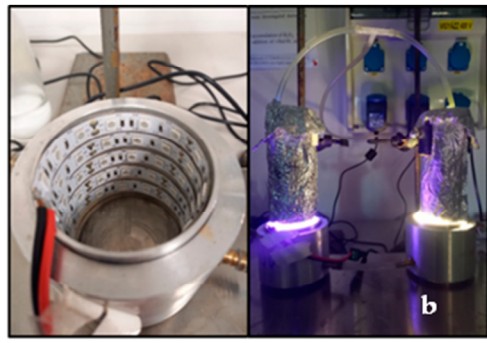
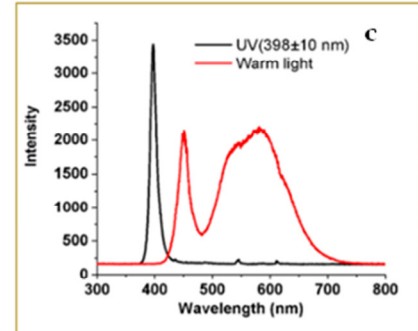

**Figure 11.** The schematic figure (**a**) and photo (**b**) of the reactors and the emission spectra of the LED light sources (**c**).

In each case, 200 cm$^3$ suspension was irradiated in a cylindrical glass reactor (inner diameter: 40 mm) placed in the center of the double-walled reactor. The suspensions were stirred in the dark for 30 min to determine the amount of adsorbed target substance. $O_2$ or $N_2$ gas was continuously bubbled through the solution or suspension to keep constants the dissolved $O_2$ concentration; the bubbling was started 15 min before starting the reaction. The experiments were started by turning on the light source and adding PDS solution to the suspension containing photocatalysts and organic target substance. The initial concentration of HQ, BQ, SMP, and TRIM was $5.0 \times 10^{-5}$ M or $1.0 \times 10^{-4}$ M, while the BiOI photocatalyst dosage was 1.0 g dm$^{-3}$. The concentration of PDS was changed within the range of 0.2–$2.0 \times 10^{-3}$ M.

After sampling, 25 μL 0.3 M $Na_2S_2O_3$ solution was added to the sample to decompose the remaining PDS. The photocatalyst samples were centrifuged immediately (Dragonlab, 15,000 RPM) and filtered with syringe filters (0.22 μm, FilterBiO, PVDF-L) before further analysis.

### 3.4. Characterization of the Light Sources

The emission spectra of the LEDs (Figure 11c) were recorded using a two-channel fiber-optic CCD spectrometer (AvaSpec-FT2048) in the 180–880 nm wavelength range. The photon flux of the light sources was determined using Reinecke's salt [75,76] and the widely applied ferrioxalate [77,78] actinometry. Reinecke's salt actinometry can be applied in the visible range, in the 400–600 nm region, while ferrioxalate actinometry can be used in the UV and near-UV regions (254–500 nm) [78].

The UV LED's photon flux determined by Reinecke's salt actinometry was $(5.81 \pm 0.03) \times 10^{-6}$ mol$_{\text{photon}}$ s$^{-1}$, and a slightly lower value, $(5.12 \pm 0.02) \times 10^{-6}$ mol$_{\text{photon}}$ s$^{-1}$ was obtained by ferrioxalate actinometry [77]. The photon flux was $(3.25 \pm 0.25) \times 10^{-6}$ mol$_{\text{photon}}$ s$^{-1}$ for the Vis-LEDs, obtained by Reinecke's salt actinometry. For calculating the apparent quantum yield of the transformation, the photon flux determined by Reinecke's salt actinometry was applied.

### 3.5. Characterization of Photocatalysts

The synthesized catalysts were characterized using powder X-ray diffractometry (XRD) (Rigaku Miniflex II, Cu Kα radiation source, 5.0–90.0 2Theta° range, with 4.0 2Theta° min$^{-1}$ resolution). The surface area was determined via $N_2$ adsorption/desorption isotherms using a Quantachrome NOVA 2200 analyzer. Diffuse reflectance spectroscopy (DRS) was

performed using an Ocean Optics DH-2000 light source and Ocean Optics USB4000 detector. The band gap energy values were evaluated by the Kubelka-Munk approach and the Tauc plot [79] and reinforced by the first derivative approach method [80].

### 3.6. Analytical Methods

HPLC measurements were performed with an Agilent 1100 HPLC equipped with a diode array UV detector (DAD) to separate the intermediates and determine 1,4-HQ, 1,4-BQ, SMP, and TRIM concentration in the treated suspension. The stationary phase was a Kinetex 2.6u XB-C18 100A (Phenomenex) reverse phase column, and the mobile phase consisted of 10 $v/v$% acetonitrile and 90% formic acid solution (0.1 $v/v$%) for the analysis of SMP containing samples. The flow rate of eluent was 0.70 mL min$^{-1}$. The same column was used for BQ and HQ containing samples; the mobile phase consisted of 50% methanol and 50% water. The flow rate of eluent was 1.0 mL min$^{-1}$. To analyze TRIM-containing solutions, Gemini 3u C6-phenyl 110A reverse phase column was used with 10% methanol and 90% ammonium-formate buffer (20 mM, pH = 3.7) as eluent; the flow rate was 0.4 mL min$^{-1}$. The detection wavelength was 250, 290, 261, and 275 nm for BQ, HQ, SMP, and TRIM, respectively. The transformation efficiency of target substances was characterized by the initial degradation rate and percent of decomposed material after 10 and 60 min treatment. The initial transformation rate was obtained from linear regression fit to the concentration-time plot, generally up to 20% transformation. Some experiments were repeated three times to check the reproducibility of the experimental results.

The transformation of PDS was followed by the determination of $SO_4^{2-}$. The $SO_4^{2-}$ concentration was measured using ion chromatography (Shimadzu Prominence LC-20AD). Shodex 5U-YS-50 column for cation detection (eluent contained 4.0 mM methanesulfonic acid and 2.5 mM phthalic acid), and Shodex NI-424 5U for anion detection (eluent: 2.3 mM aminomethane solution). The flow rate of the mobile phase was 1.0 mL min$^{-1}$.

The total organic carbon (TOC) content of the samples was determined by a Multi N/C 3100 analyzer (Analytik Jena, Jena, Germany) equipped with an NDIR detector. For spectrophotometric measurements, an Agilent 8453 UV-Vis spectrophotometer was used.

### 4. Conclusions

In this work, the effect of PDS on the transformation of various organic substances was investigated, in BiOCl, BiOBr, and BiOI suspensions, under UV (398 nm) and Vis radiation. The method described in the literature was used for the synthesis of the BiOX catalysts; their modification was not aimed. The parameters of the catalysts corresponded to the literature data; accordingly, 398 nm UV light is theoretically not suitable for the excitation of BiOCl; BiOBr can be excited mainly by UV light, while BiOI can also be efficiently excited by visible light. Due to PDS being an effective $e_{cb}^-$ scavenger and the source of sulfate radical ion ($SO_4^{\bullet-}$), an enhanced transformation rate of organic compounds was expected.

Without organic matter, BiOI showed outstanding activity in PDS transformation; 15–20% of PDS was transformed even under Vis radiation, while for BiOBr and BiOCl only 3–4% of PDS transformed under 398 nm irradiation. The increased efficiency of HQ transformation in the presence of PDS can be partially attributed to PDS activation by HBQ$^\bullet$-this process contributed dominantly to the activation of PDS and the increase of HQ conversion for BiOCl and BiOBr, and additively for BiOI. BiOI showed the highest activity, especially under visible light radiation, most probably due to its outstanding activity in the activation of PDS among the tested BiOXs. For BiOBr and BiOCl, 75% conversion was reached during 60 min even under UV radiation in the presence of PDS. The efficiency was far exceeded by BiOI, even under Vis radiation; the HQ wholly transformed within 20 min. Similarly, for BiOI the SMP conversion increased to more than 90% from 40%. However, the conversion efficiency of SMP was still much lower than that of HQ when the HBQ$^\bullet$-initiated process also contributed to the enhanced efficiency.

Our results reflect that the activity of BiOX catalysts, through various factors, significantly depends on the model compound used-both in the presence and without PDS. For

BiOCl and BiOBr, most likely, photogenerated oxygen-vacancies and $Bi^0$ play an important role and influence the activity. In $O_2$-free, PDS-containing suspensions, the efficient transformation of HQ and SMP confirmed that the improved activity is primarily related to the enhanced efficiency of direct charge transfer and $SO_4^{\bullet-}$ formation.

The structure and activity of BiOI did not change during three consecutive cycles in the presence of PDS; however, in the biologically treated wastewater, the organic compounds of the matrix adsorbed on the BiOI surface blocked the processes taking place and significantly reduced the efficiency. Furthermore, BiOI was sensitive to $HCO_3^-$.

Our results proved that combining the PDS process with BiOI photocatalyst is a promising process that can be used for the selective conversion and degradation of organic pollutants since its effectiveness depends significantly on the properties of the compound to be removed. Some critical limiting factors in terms of efficiency and applicability are highlighted. In addition, many questions arose relating to the processes taking place on the surface of BiOX catalysts, as well as the formation and role of different reactive species, which motivate further research to obtain a more comprehensive picture of the activation of PDS with BiOX photocatalyst.

**Supplementary Materials:** The following supporting information can be downloaded at: https://www.mdpi.com/article/10.3390/catal13030513/s1, Table S1. The parameters of the biologically treated domestic wastewater; Table S2. The name, manufacturer, and purity of the used materials; Figure S1. The transformation of TRIM in aerated BiOI suspensions, irradiated with Vis LED ($c_0$(TRIM) = 1.0 × 10−4 M; $c_0$(SMP) = 1.0 × 10−4 M; NaF was added for desorption); Figure S2. The Effect of PDS, inorganic ions, and SMP on the XRD diffractogram of BiOI photocatalysts under UV and Vis radiation in aerated suspensions; Figure S3. The Effect of PDS and inorganic ions on the DRS spectra diffractogram of BiOI photocatalysts under UV and Vis radiation in aerated suspensions; Figure S4. The effect of pH and inorganic ions on the color of BiOI suspension (a: Milli-Q; b: biologically treated domestic wastewater, pH = 7.8; c: biologically treated domestic wastewater, pH = 4; d: $HCO_3^-$ solution; e: the pH of the $HCO_3^-$solution was adjusted to 4 and stirred for 10 min. before suspending the BiOI).

**Author Contributions:** Conceptualization, writing—original draft preparation, review and editing, funding acquisition: T.A.; investigation: B.V., M.N. and L.F.; XRD and DRS measurements and data evaluation: Z.P.; visualization and TOC measurement, A.C. All authors have read and agreed to the published version of the manuscript.

**Funding:** This work was financed by the National Research, Development, and Innovation Office—NKFI Fund OTKA, project number FK132742. Tünde Alapi, thanks for the support of the János Bolyai Research Scholarship of the Hungarian Academy of Sciences. Luca Farkas, thanks for the financial support of the ÚNKP-22-3-SZTE-398 New National Excellence Program of the Ministry for Culture and Innovation from the source of the National Research, Development, and Innovation Fund.

**Data Availability Statement:** The data is included in the article or Supplementary Material.

**Acknowledgments:** The authors thank the Szegedi Vízmű Zrt. for providing the biologically treated domestic wastewater as a matrix.

**Conflicts of Interest:** The funders had no role in the design of the study; in the collection, analyses, or interpretation of data; in the writing of the manuscript; or in the decision to publish the results.

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
