# Peer review of "Application of BiOX Photocatalyst to Activate Peroxydisulfate Ion-Investigation of a Combined Process for the Removal of Organic Pollutants from Water"

_catalysts, doi:10.3390/catal13030513_

Round 1

Reviewer 1 Report

The following issues need to be addressed before the manuscript could be accepted:

1. The manuscript mainly focused on the transformation efficiencies of different organic compounds. The novelty is not clear. More contents should be added on the catalyst characterization and catalytic mechanisms to increase the novelty.

2. In Fig. 2, there is no Fig. 2f. The “e” should be “f”. Similar problem in Fig. 3.

3. In Fig, 3, how about the photolysis of HQ?

4. Line 170-174, the transformation of HQ was due to the reaction with O2•−. However, there is no evidence. More evidence for the existence of O2•− should be provided, such as EPR.

5. The pH is closely related to the PDS activation. The pH changes should be monitored during the reactions.

6. What are the mineralization efficiencies for the treated organic compounds? Has the authors test the total organic carbon(TOC)?

7. What are the morphology, particle size and BET surface areas of the BiOX? These are important for discussion of the catalytic performances of the BiOX.

Author Response

Reviewer 1.

Thank you for the reviewer's valuable work and comments. The corrections and changes are highlighted in the manuscript.

  1. The manuscript mainly focused on the transformation efficiencies of different organic compounds. The novelty is not clear. More contents should be added on the catalyst characterization and catalytic mechanisms to increase the novelty.

The novelty of the manuscript is the investigation and comparison of the PDS addition on the efficiency of BiOI, BiOBr, and BiOCl photocatalysts using four different organic substances, including the investigation the efficiency of PDS transformation (which is very limited in the literature) and investigation the reusability of BiOI and matrix effect on the BiOI activity. To the best of our knowledge, BiOX catalysts have not been investigated and compared to this extent from the point of view of combination with PDS. The strength of the manuscript is that we chose four systematically selected target compounds from multiple, well-defined points of view and tested the effect of PDS under visible and 398 nm radiation. The manuscript also covers the reusability of BiOI and the examination of the matrix effect and some critical limiting factors in terms of efficiency and practicality are highlighted.

For preparation of BiOX photocatalysts we used a published method described by Bardos et al.([15] Bárdos, E.; Király, A.K.; Pap, Z.; Baia, L.; Garg, S.; Hernádi, K. The Effect of the Synthesis Temperature and Duration on the Morphology and Photocatalytic Activity of BiOX (X = Cl, Br, I) Materials. Appl. Surf. Sci. 2019, 479, 745–756, doi:10.1016/j.apsusc.2019.02.136). Our work did not aim to develop the synthesis method. The detailed characterization of BiOI, BiOBr and BiOCl synthesized by the same process was done in the mentioned manuscript; we did not want to repeat the data. The most important parameters were determined (XRD, DRS measurements and determination of specific surface area has done), but the morphology was not investigated. According to the publication of Bardos et al., the BiOX showed spherical hierarchical microcrystals composed from individual nanoplates. The manuscript was completed to clarify the aim of our work and to provide information about the morphology of the BiOX photocatalysts.

  1. In Fig. 2, there is no Fig. 2f. The “e” should be “f”. Similar problem in Fig. 3.

Thank you, the figure is corrected.

  1. In Fig, 3, how about the photolysis of HQ?

There is no transformation of HQ via direct photolysis. It was measured and described in the text: 176-178: „The concentration of the HQ does not change in irradiated and aerated solutions or O2-free suspensions and is poorly (<2%) adsorbed on the surface of photocatalysts.”

  1. Line 170-174, the transformation of HQ was due to the reaction with O2•−. However, there is no evidence. More evidence for the existence of O2•− should be provided, such as EPR.

As reviewers know well, in the case of Catalysts, authors have 10 days to change the manuscript, which is not enough to organize EPR measurements. We agree that the formation of individual radicals can be clarified with EPR measurements, but based on the literature, we did not consider this necessary in this case. The referred articles support that, in the case of heterogeneous photocatalysis, the transformation of HQ is primarily due to the reaction with O2•−, even in the case of TiO2 when hydroxyl radical formation is characteristic besides O2•− (doi:10.1016/j.jphotochem.2020.113057). The importance of O2•− is more evident in the case of BiOX photocatalysts, when hydroxyl radical formation is neglected, even for BiOCl and BiOBr. as it was reported in high quality publications (cited in the manuscript). The formation of O2•− in the case of BiOX photocatalysts, as the main reactive species, was already confirmed by others.

In another article, written by the corresponding author and published in Chemosphere (doi:10.1016/j.chemosphere.2021.130636) the same procedure was used to compare the O2•− formation rates in the case of BiOI/BiOCl composite catalysts.

  1. The pH is closely related to the PDS activation. The pH changes should be monitored during the reactions.

I agree with the reviewer that pH can affect the efficiency of the heterogeneous photocatalysis in several ways, such as protonation-deprotonation of organic substances affecting their reactivity towards radicals; also, the surface properties can change, and by this way, the adsorption properties of the photocatalyst. In the case of the PDS processes, the SO4•− can participate in a pH-dependent reaction to produce hydroxyl radicals, which is predominant in alkaline media, especially above pH 9. Recently, the role of pH in the decomposition of persulfate in aqueous solutions was investigated by Jorge Herrera-Ordonez (https://doi.org/10.1016/j.ceja.2022.100331), and pH did not have a significant effect in the range of pH 3 -9.

In the case of our work, the pH of BiOX suspension was about 6.0 - 6.5, the addition of HQ, BQ, SMP or TRIM did not significantly change the pH of the suspension. During treatment, pH decreased to 5.0 – 4.0 without PDS, and to 4.5 – 3.7 in the presence of PDS. Most probably, the change in pH has no significant effect on the PDS transformation and SO4•− formation; however, the surface properties of the photocatalysts and the ratio of HO2/O2•− can change during treatment. As was described in the manuscript (2.7. Effect of inorganic ions and biologically treated domestic wastewater as matrix), the effect of pH was investigated at pH 4, but a significant change in the efficiency was not observed (Fig. 10): “The change of pH to 4 with sulfuric acid caused a slight decrease in the transformation rate compared to that determined in Milli-Q water.” (Line

The manuscript was completed with data on the change in pH and its possible effects on the process. (Line 416-417)

  1. What are the mineralization efficiencies for the treated organic compounds? Has the authors test the total organic carbon(TOC)?

Although SO4•− is effective for mineralization, significant TOC changes were not observed during the treatments. TOC did not decrease during the 60-minute treatment without PDS, while a maximum decrease of 10% was measured in the presence of PDS. The manuscript was completed with these data.

The text was completed: “Although SO4•− is effective for mineralization, significant TOC changes were not observed during the treatments. TOC did not decrease during the 60 minute treatment without PDS, while a maximum decrease of 10% was measured in the presence of PDS for HQ and SMP.” Line 333-335

  1. What are the morphology, particle size and BET surface areas of the BiOX? These are important for discussion of the catalytic performances of the BiOX.

The characterization of BiOX photocatalysts are summarized in Table 1. and in Fig 1. The Table 1. was completed with primary particle size. As mentioned in the Chapter “4.2. Preparation of BiOX photocatalysts” we used a publish method described by Bardos et al.:

[15] Bárdos, E.; Király, A.K.; Pap, Z.; Baia, L.; Garg, S.; Hernádi, K. The Effect of the Synthesis Temperature and Duration on the Morphology and Photocatalytic Activity of BiOX (X = Cl, Br, I) Materials. Appl. Surf. Sci. 2019, 479, 745–756, doi:10.1016/j.apsusc.2019.02.136. Our work did not aim to develop the synthesis method. The detailed characterization of BiOI, BiOBr and BiOCl synthesized by the same process was done in the mentioned manuscript; we did not want to repeat the data. The most important parameters were determined (XRD, DRS measurements and determination of specific surface area has done), but the morphology was not investigated. According to the publication of Bardos et al., the BiOX showed spherical hierarchical microcrystals composed from individual nanoplates. The manuscript was completed to clarify the aim of our work and to provide information about the morphology of the BiOX photocatalysts.

Reviewer 2 Report

(1)      Pls provide full name of “PDS” in title and “HQ” (line 17) for their appearance.

(2)      Pls unified writing style of “organic pollutants/organic contaminants/organic substances” and “peroxydisulfate/peroxodisulfate”.

(3)      Pls add “BiOX” into keywords.

(4)      Pls provide the typical concentration range of antibiotics in wastewater in line 36.

(5)      Some relevant references, such as 10.1016/j.cej.2020.128176, 10.1016/j.chemosphere.2016.02.089, 10.1016/j.cclet.2021.10.087 should be cited in lines 77-86 in introduction.

(6)      Pls labeled sub a/b/c in Figure 1 and related Figures as Figure 2 did.

(7)      Figure 8: pls correct the writing style of O2 and N2.

Author Response

Reviewer 2

Thank you for the reviewer's valuable work and comments. The corrections and changes are highlighted in the manuscript.

  • Pls provide full name of “PDS” in title and “HQ” (line 17) for their appearance.
  • Pls unified writing style of “organic pollutants/organic contaminants/organic substances” and “peroxydisulfate/peroxodisulfate”.

We have corrected the manuscript.

(3)      Pls add “BiOX” into keywords.

The title of the manuscript includes "BiOX"; that was why we didn't use it as a keyword. But at the reviewer's suggestion, we included it among the keywords.

(4)      Pls provide the typical concentration range of antibiotics in wastewater in line 36.

The manuscript was completed: „The concentration of antibiotics in influent wastewater is typically in the order of nanograms/liter and often reaches several milligrams per liter. During wastewater treatment, their concentration is usually reduced, although a significant part of them and some of their metabolites can often remain in the effluent and reach the surface waters.”

(5)      Some relevant references, such as 10.1016/j.cej.2020.128176, 10.1016/j.chemosphere.2016.02.089, 10.1016/j.cclet.2021.10.087 should be cited in lines 77-86 in introduction.

The introduction was completed with the suggested publications (Line 91-93)

10.1016/j.cej.2020.128176: Enhanced removal of methylparaben mediated by cobalt/carbon nanotubes (Co/CNTs) activated peroxymonosulfate in chloride-containing water: Reaction kinetics, mechanisms and pathways

10.1016/j.chemosphere.2016.02.089 Activation of peroxymonosulfate by base: Implications for the degradation of organic pollutants

10.1016/j.cclet.2021.10.087 Enhanced degradation of organic contaminants by Fe(III)/peroxymonosulfate process with l-cysteine

(6)      Pls labeled sub a/b/c in Figure 1 and related Figures as Figure 2 did.

(7)      Figure 8: pls correct the writing style of O2 and N2.

Figures are checked and corrected.

Reviewer 3 Report

This paper describes the synthesis of BiOX and its application in the removal of organic pollutants from water.

1.        The manuscript has scientific soundness but requires revision. The authors should revise the introduction part by stating the importance of BiOX. Why authors selected these compounds for photocatalytic degradation of pollutants in water?

2.        The authors should highlight in the manuscript the reasons for higher efficiencies of BiOI.

3.        Abstract and conclusion must be rewritten by mentioning the actual data. The authors should describe well why BiOI showed higher efficiency than others.

4.        The manuscript requires extensive English revision.

5.        The manuscript has many errors and missing letters. The manuscript should be revised for all mistakes.

6.        The authors should reference important papers published in the past 3-5 years such as.

https://doi.org/10.1007/s13762-022-04216-6

DOI: 10.1039/D1RA04649G

https://doi.org/10.1016/j.jenvman.2021.112605

Author Response

Reviewer3

Thank you for the reviewer's valuable work and comments. The corrections and changes are highlighted in the manuscript.

The manuscript has scientific soundness but requires revision. The authors should revise the introduction part by stating the importance of BiOX.

The introduction (line 55-71) contains information about BiOX and describe the importance of BiOX photocatalysts. The manuscript is related to three topics: the problem caused by environmentally relevant organic pollutants, the application of BiOX photocatalysts to remove them and the PDS-based processes. We think that, it is a matter of taste which topics the introduction starts with, especially if all three are discussed sufficiently. Since only one of the three reviewers requested the revision of the introduction's structure, we thought it was acceptable in its original form, so we did not change it.

Why authors selected these compounds for photocatalytic degradation of pollutants in water?

The organic compounds, depending on their structure, are reactive to different degrees towards various reactive species. With their appropriate selection as target substance, the formation and contribution of reactive species to thee transformations can be assessed. The target compounds were chosen to characterize the activity of catalysts from different points of view:

  • 1,4-BQ transformation in O2-free suspension was investigated because of the amount of the formed HQ is proportional to that of ecb− on the surface of the irradiated photocatalyst (using a given photon flux).
  • 1,-4-HQ transformation in O2-containing suspensions was studied to compare the formation rate of O2•− under visible and UV radiation for each BiOX catalysts. The transformation of HQ is primarily due to the reaction with O2•−, (doi:10.1016/j.jphotochem.2020.113057). The importance of O2•− is evident in the case of BiOX photocatalysts. In another article, written by the corresponding author and published in Chemosphere (doi:10.1016/j.chemosphere.2021.130636) the same procedure was used to compare the O2•− formation rates in the case of BiOI/BiOCl composite catalysts.
  • SMP and TRIM, as widely used, hardly biodegradable antibiotics, often detected in surface water, were used as environmentally relevant target substances. A significant difference between the substrates is that SMP is poorly adsorbed on BiOX photocata-lysts, while the adsorption of TRIM is substantial, particularly in the case of BiOI. Thus, in this way the importance of adsorption was also studied.

The aspects listed here are also described in the manuscript.

  1. The authors should highlight in the manuscript the reasons for higher efficiencies of BiOI.

The possible reasons for the higher efficiency of BiOI are discussed in the manuscript (Chapter 2.3.1.) and highlighted in Conclusion (Line 521 and Line 528-530).

Although possible reasons for the observed superior activity of BiOI in the presence of PDS have been described, further exact interpretations are beyond the scope and purpose of this manuscript and require further investigation, as noted in the original manuscript.

  1. Abstract and conclusion must be rewritten by mentioning the actual data. The authors should describe well why BiOI showed higher efficiency than others.

The abstract and conclusions have been partially modified according to the comment.

  1. The manuscript requires extensive English revision. The manuscript has many errors and missing letters. The manuscript should be revised for all mistakes.

We have reviewed and corrected the manuscript to the best of our knowledge, and English revision was made.

  1. The authors should reference important papers published in the past 3-5 years such as.

https://doi.org/10.1007/s13762-022-04216-6, DOI: 10.1039/D1RA04649G, https://doi.org/10.1016/j.jenvman.2021.112605

The proposed publications:

  • Enhanced photo-Fenton degradation of Rhodamine B using iodine-doped iron tungstate nanocomposite under sunlight (International journal of Environmental Science and Technology)
  • Efficient Cr(vi) photoreduction under natural solar irradiation using a novel step-scheme ZnS/SnIn4S8 nano heterostructured photocatalysts (RSC Advances)
  • Subcritical and supercritical water oxidation for dye decomposition (Journal of Environmental Management)

are not related closely to the topic of the manuscript, so we do not refer to them. Let me draw the reviewer's attention to the fact that most of the literature used in the manuscript has been published in high-quality journals closely related to the topic of the manuscript within the last years.

Round 2

Reviewer 1 Report

The manuscript has been improved significantly and can be published in the current form. 

Reviewer 3 Report

All the reviewer's concerns were answered.